# The level of postoperative care influences mortality prediction by the POSPOM score: A retrospective cohort analysis

Jan Menzenbach[1]☉, Yannik C. Layer[1]☉, Yonah L. Layer[1], Andreas Mayr[2], Mark Coburn[1], Maria Wittmann[1‡], Tobias Hilbert[1‡]*

**1** Department of Anesthesiology and Intensive Care Medicine, University Hospital Bonn, Bonn, Germany,
**2** Institute of Medical Biometrics, Informatics and Epidemiology (IMBIE), University Hospital Bonn, Bonn, Germany

☉ These authors contributed equally to this work.
‡ These authors also contributed equally to this work
* thilbert@uni-bonn.de

## Abstract

### Background

The Preoperative Score to Predict Postoperative Mortality (POSPOM) assesses the patients' individual risk for postsurgical intrahospital death based on preoperative parameters. We hypothesized that mortality predicted by the POSPOM varies depending on the level of postoperative care.

### Methods

All patients age over 18 years undergoing inpatient surgery or interventions involving anesthesia at a German university hospital between January 2006, and December 2017, were assessed for eligibility for this retrospective study. Endpoint was death in hospital following surgery. Adaptation of the POSPOM to the German coding system was performed as previously described. The whole cohort was divided according to the level of postoperative care (normal ward vs. intensive care unit (ICU) admission within 24 h vs. later than 24 h, respectively).

### Results

199,258 patients were finally included. Observed intrahospital mortality was 2.0% (4,053 deaths). 9.6% of patients were transferred to ICU following surgery, and mortality of those patients was increased already at low POSPOM values of 15. 17,165 patients were admitted to ICU within 24 h, and these patients were older, had more comorbidities, or underwent more invasive surgery, reflected by a higher median POSPOM score compared to the normal-ward group (29 vs. 17, p <0.001). Mortality in that cohort was significantly increased to 8.7% (p <0.001). 2,043 patients were admitted to ICU later than 24 h following surgery (therefore denoted unscheduled admission), and the median POSPOM value of that group

**Data Availability Statement:** All relevant data sets necessary to replicate the study findings are within the paper and its Supporting information files.

**Funding:** The authors received no specific funding for this work.

**Competing interests:** The authors have declared that no competing interests exist.

was 23. Observed mortality in this cohort was highest (13.5%, p <0.001 vs. ICU admission <24 h cohort).

## Conclusion

Increased mortality in patients transferred to high-care wards reflects the significance of, e.g., intra- or early postoperative events for the patients' outcome. Therefore, scoring systems considering only preoperative variables such as the POSPOM reveal limitations to predict the individual benefit of postoperative ICU admission.

## Introduction

Precise assessment of the surgical patient's individual risk is the pivotal issue of preoperative anesthesiologic evaluation. It is essential for clinical decision-making and to assure that perioperative mortality, which has steadily decreased over the decades, may continuously be controlled despite increasing patient age. Global prevalence of postoperative mortality is heterogeneous. It is estimated that 4.2 million patients per year die within 30 days after surgery, accounting for almost 8% of all annual deaths worldwide [1,2]. Given a steadily increasing number of surgical interventions, rising from 226 million in 2004 by well over a third to 313 million within eight years [3] and conservatively assuming only a constant growth rate, over 410 million surgical interventions worldwide can be expected in 2021 on a cautious estimate. This underlines the significance of a precise but also time-effective risk assessment.

Scoring systems provide assistance in objectifying the patient's individual risk. This is, first of all, determined by his or her comorbidities and physiological reserves. The worldwide frequently used American Society of Anesthesiologists (ASA) physical status score is based on the anesthesiologist's subjective assessment and categorizes patients into five risk groups [4]. This considerably rough gradation, the subjectivity as well as the fact that age, which was demonstrated to significantly influence perioperative risk [5], is ignored when calculating the ASA score, limit its accuracy and validity [6,7]. Other scoring systems such as the Physiological and Operative Severity Score for the enUmeration of Mortality and Morbidity (POSSUM) also take into account the various categories of surgical interventions and the invasiveness, which likewise determine perioperative morbidity and mortality [8,9]. However, as intra- as well as postoperative details are required, the POSSUM is not appropriate for preoperative risk assessment.

In 2016, Le Manach et al. presented the Preoperative Score to Predict Postoperative Mortality (POSPOM) [10]. All patients undergoing surgery or otherwise interventions involving anesthesia performed in France during one year were retrospectively analyzed, including the patients' comorbidities and age as well as the type of intervention, and assigned to the individual postoperative intrahospital mortality. This resulted in a scale of POSPOM values, indicating the patient's individual risk for postoperative intrahospital death derived from objective preoperative parameters. The POSPOM obtains its predictive power not at least from the remarkable size of the cohort used for its derivation comprising over 2.7 million patients. Its discriminative strength has been successfully demonstrated in patient populations undergoing abdominal, vascular, and orthopedic surgery in countries outside France [11–15].

The level of postoperative care (normal ward or intensive care unit [ICU]) may exert influence on intrahospital mortality [16]. However, this level was ignored when Le Manach et al. derived and validated the POSPOM. We hypothesized that the normalized predicted

postoperative mortality according to the POSPOM varies depending on whether patients are transferred either to the normal ward or the ICU following surgery, thus impairing the predictive validity of the score. To test our hypothesis, we used a POSPOM adapted from the French to the German coding system as previously described [15].

## Methods

All analyses were performed in accordance with the Declaration of Helsinki. The local ethics committee (University Hospital Bonn, Bonn, Germany) considered the retrospective study to be compliant with the terms of the current professional codes and regulations and thereby approved the study protocol. Due to its retrospective character, written informed consent was waived. The work has been reported in line with the STROCSS criteria [17]. Adaptation of the POSPOM scoring system to the German coding system was performed as previously described [15]. In brief, all data were extracted from the anonymized electronic database generated as provided by law following the German hospital fees act (*Krankenhausentgeltgesetz* [KHEntG]), §21 [18]. All patients age over 18 years undergoing inpatient surgery or interventions involving anesthesia at the German University Hospital Bonn between January 1, 2006, and December 31, 2017, were considered eligible for this study. The disciplines and procedures covered by the observation comprised cardiac, urologic, vascular, plastic, ophthalmologic, gynecologic, ear-nose-throat (ENT), orthopedic, transplant, digestive, liver, biliary tract and pancreas, thoracic and neurosurgery (see S1 Table). Furthermore, other interventions involving anesthesia including endoscopy, neuroradiology and cardiac rhythmology were likewise included. Primary endpoint was death in hospital following surgery. Death after hospital discharge was not taken into account. The French codes for the classification of medical procedures *Classification Commune des Actes Médicaux* (CCAM) were assigned to their German equivalent (*Operationen- und Prozedurenschlüssel* [OPS]) [19]. The surgical procedure used for the analysis is the index operation, reflecting the scheduled surgical procedure at admission. For patients with multiple surgical procedures during the same stay, the index procedure was defined as the first one performed during the stay. In case of a patient having more than one relevant surgery encoded at the same time, we assigned the surgery scoring the most POSPOM points. Comorbidities were encoded using the International Classification of Diseases (10[th] revision [ICD-10]) disease categorization [20]. All variables included in the score are given in S1 Table.

The whole patient cohort was divided according to the level of postoperative care following the index surgery (normal ward vs. ICU). By definition, the normal-ward cohort comprised patients that had never been admitted to the ICU during their postoperative hospital stay. The ICU cohort was furthermore split according to the time of admission following the index surgery (within 24 h or later than 24 h, respectively).

The observed postoperative mortality reported by Le Manach et al. for their validation cohort ($m_{vc}$) was used as reference to compare our results with. Data for this cohort were extracted from the original publication [10]. Since the authors do not report $m_{vc}$ results for POSPOM values <1 and >40, patients with these scoring results had to be excluded from the analysis.

All analyses were performed using R (Version 3.5.0 [http://www.r-project.org], last date accessed: January 23, 2020) under creative common license, and affiliated packages (ggplot2, dplyr, and pROC). Further statistical analyses and visualization were performed using MS Excel 2019 (Microsoft Corp., Redmond, CA, USA) and GraphPad PRISM 8 (La Jolla, CA, USA). All data are presented as absolute numbers or, as indicated, as mean or median values with standard deviation (SD) or 25-to-75 percentile, respectively. Significance of differences in median POSPOM values between groups was calculated using the Mann-Whitney U test, differences in mortality were calculated using the Chi square test.

All relevant data sets necessary to replicate the study findings are within the paper and its Supporting Information files.

## Results

A total of 357,861 patients were identified from the data base during the observation period from January 2006 to December 2017. 158,081 patients were excluded as they did not meet eligibility criteria. These cases involved 115,281 procedures without anesthesia, 41,836 patients younger than 18 years and 964 cases with missing data. Furthermore, 522 patients with POSPOM values <1 or >40 were excluded (see Methods section). The remaining 199,258 patients were included in the final analysis (Fig 1). Adaptation of the French POSPOM scoring model to the German coding system was performed as previously described [15]. The distribution of POSPOM score values across the surgical disciplines as well as the characteristics of patient age and comorbidities are shown in detail in the Supplemental S2 and S3 Tables.

180,050 patients (90.4%) have not ever been transferred to the ICU during their hospital stay, forming the largest subgroup, the normal-ward cohort (Figs 1 and 2). 17,165 patients were transferred to the ICU within 24 h following surgery and therefore denoted scheduled admission, while 2,043 patients were admitted to ICU later than 24 h following surgery (denoted unscheduled admission), thereby forming the smallest subcohort. In the whole cohort, mean age was 56.4 years (SD 18.5), and 50.8% (101,117 patients) were of male sex. Table 1 shows details on the patient characteristics of the whole as well as of the three subcohorts.

We compared mortality in our hospital with that observed and reported for the validation cohort ($m_{vc}$) by Le Manach et al [10]. In our whole cohort, observed intrahospital mortality was 2.03% (4,053 deaths out of 199,258 cases) and therefore marginally higher (factor increase 1.18) compared to the weighed mortality reported by Le Manach et al. ($m_{vc}$ 1.72%, corresponding to 3,426 deaths). An increased mortality was particularly evident at lower POSPOM values, while at those above 30, mortality rate for our hospital actually revealed lower as described by the French group (Fig 3A, grey panel). Focusing on those patients that were not ever admitted to the ICU (normal-ward cohort), observed postoperative mortality in this group was 1.27% (2,280 deaths out of 180,050 cases), whith an $m_{vc}$ of 1.33% (corresponding to 2,388 deaths) (factor increase 0.95) (Fig 3A, green panel). As expected, patients that were transferred to the ICU within 24 h following surgery (scheduled admission) were, in general, older, had more severe comorbidities, or underwent more invasive surgical procedures than those transferred to and remaining on the normal ward following surgery. This was reflected by a significantly higher median POSPOM score of this cohort, compared to the normal-ward group (29 [25-to-75 percentile: 24 to 32] vs. 17 [11 to 23], Mann-Whitney U test p <0.001) (Fig 2, green and blue panel). Mortality compared to the normal-ward group was significantly increased, as 8.72% died in hospital following surgery in that cohort (1,497 deaths out of 17,165 cases, Chi square test p <0.001). According to the data of Le Manach et al., a mortality rate ($m_{vc}$) of 5.69% (corresponding to 977 deaths) would have been expected (factor increase 1.53) (Fig 3A, blue panel).

The subgroup of patients that were admitted to ICU later than 24 h following surgery and following an intermittent stay on the normal ward (unscheduled admission) showed a median POSPOM value of 23 (16 to 28) (Fig 2, red panel), corresponding to a significant left shift compared to the scheduled-admission group (Mann-Whitney U test p <0.001). Observed mortality in this cohort was highest, as 13.51% (276 deaths out of 2,043 cases) of patients died during their hospital stay (Chi square test compared to scheduled-admission group p <0.001). Applying the $m_{vc}$ to that group revealed a weighed mortality of 2.95% (corresponding to 60 deaths)

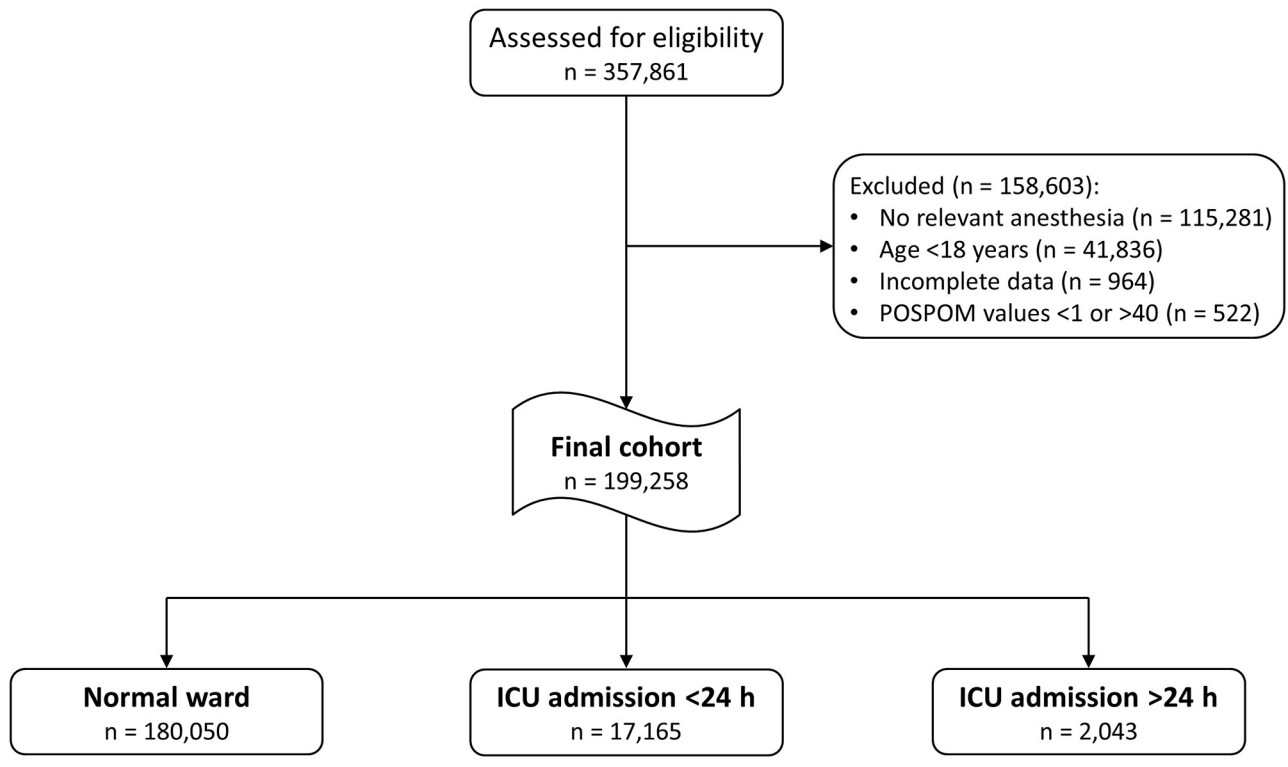

**Fig 1. Retrospective study design and patient flow chart.**

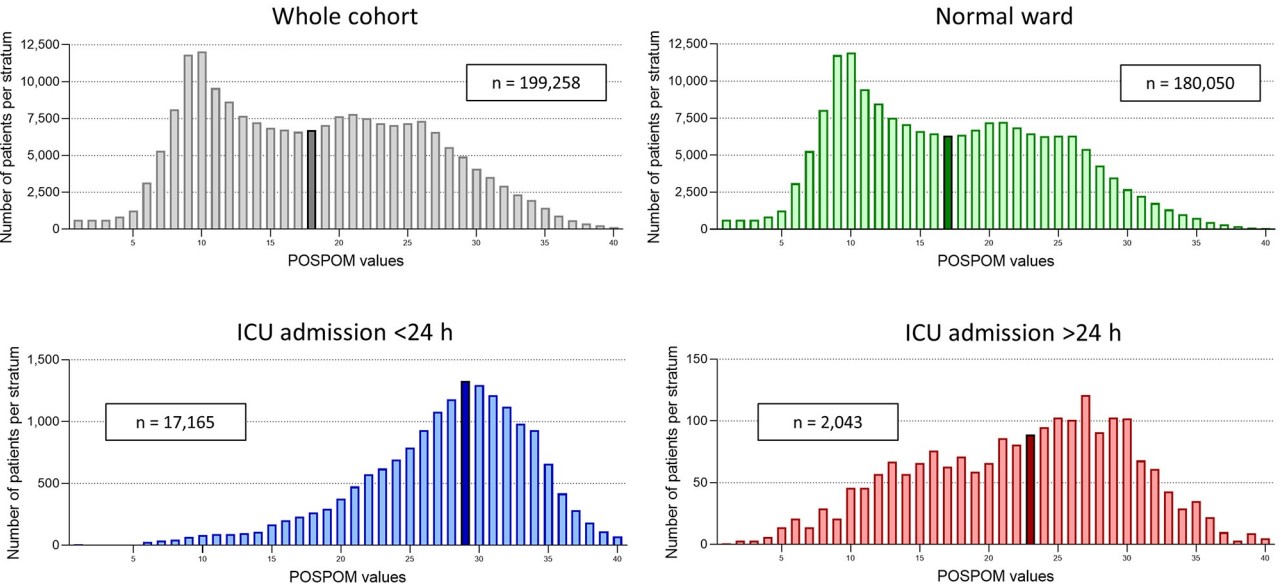

**Fig 2. Distribution of POSPOM values.** Each panel shows the distribution of POSPOM values for the respective cohort: Whole cohort (grey bars) and the subcohorts of patients that have never been transferred to the ICU during their postoperative hospital stay (termed normal-ward group; green bars) and of those that have been admitted to the ICU within 24 h (blue bars) and later than 24 h following surgery (red bars). The black-framed bar represents the median POSPOM stratum of each cohort.

**Table 1. Patient characteristics.**

|  | Whole cohort | Normal ward | ICU <24 h | ICU >24 h |
|---|---|---|---|---|
| *n (%)* | 199,258 | 180,050 (90.4) | 17,165 (8.6) | 2,043 (1.02) |
| *Age (SD)* | 56.4 (18.5) | 55.5 (18.7) | 65.2 (14.5) | 65.2 (14.7) |
| *Male n (%)* | 101,117 (50.8) | 89,150 (49.5) | 10,814 (63.0) | 1,153 (56.4) |

(factor increase 4.57). As shown in Fig 3A (red panel), the mortality we observed in that cohort exceeded the $m_{vc}$ over the whole range of POSPOM values and not only at the lower ones, in contrast to the group of patients that had never been admitted to ICU. Of note, mortality in the two subcohorts of patients that have ever been transferred to ICU (scheduled as well as unscheduled admission) exceeded that in the group of patients remaining on the normal ward by more than eight times already at low POSPOM values of 15 (Fig 3B).

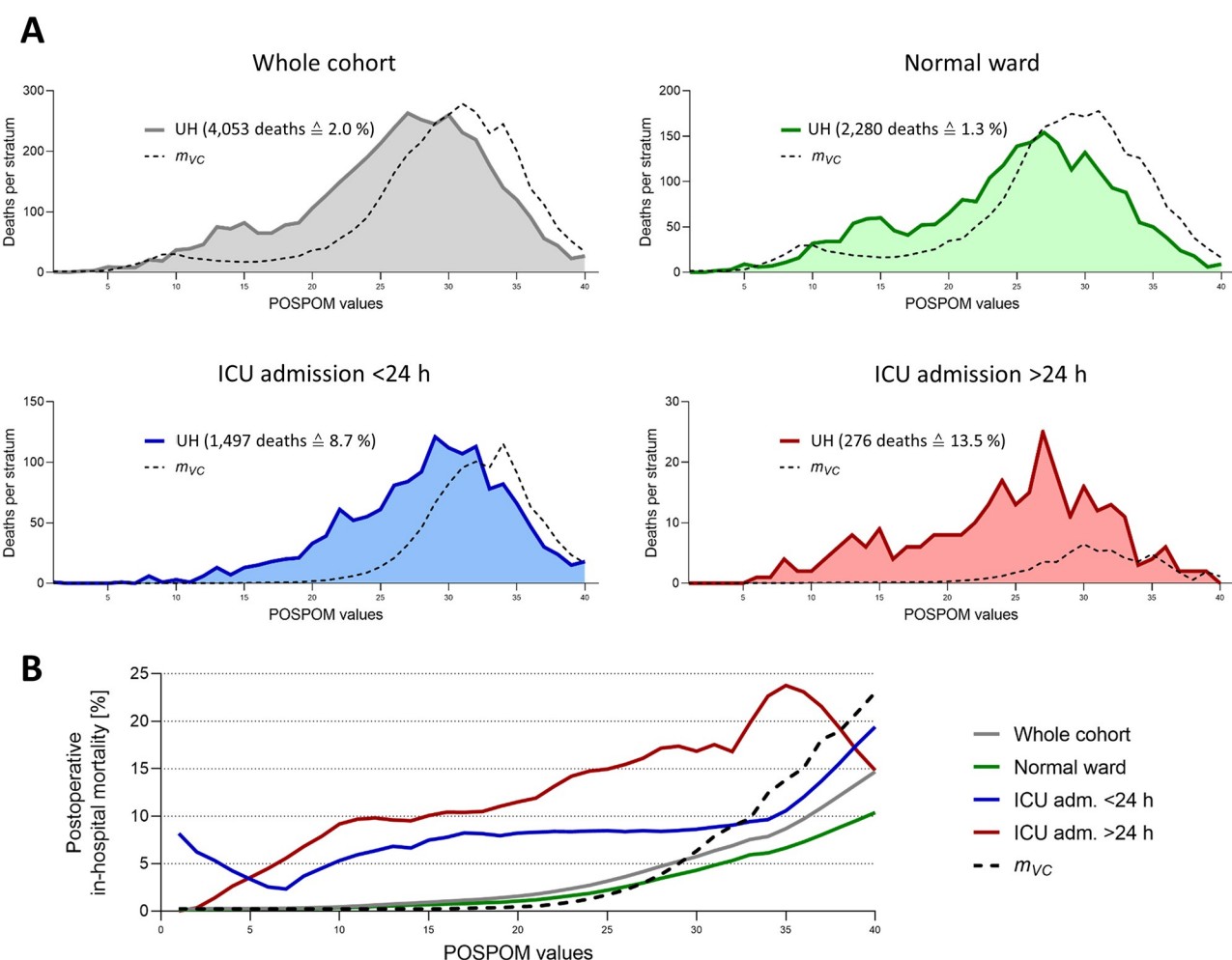

**Fig 3. Postsurgical intrahospital mortality in normal ward and ICU patients. (A)** Each panel shows the number of deaths per POSPOM stratum in the respective cohort of the study university hospital (UH), with the area under the curve representing the cumulative mortality (absolute as well as percentage numbers are given in brackets). The dashed line represents the observed postoperative mortality reported by Le Manach et al. for their validation cohort ($m_{vc}$). **(B)** Figure shows the percentage mortality for the whole cohort as well as for each of the three subcohorts (normal ward, ICU admission <24 h, ICU admission >24 h), in comparison with the observed postoperative mortality reported by Le Manach et al. ($m_{vc}$). Note that curves in Fig 3B have been smoothened.

## Discussion

The POSPOM allows to predict postoperative intrahospital mortality based on preoperative variables with remarkable predictive power. However, Le Manach et al. ignored whether patients were transferred to normal ward or to ICU following surgery when deriving and validating the score from a cohort of surgical interventions performed in France during the year 2010. After adapting the POSPOM to the German coding system, we tested the hypothesis if the level of postoperative care influences the results of the POSPOM scoring. Patients that were admitted to ICU following surgery were, in general, older, had more severe comorbidities, and underwent more invasive procedures than those transferred to the normal ward, as revealed by higher median POSPOM scores. This was particular evident for those patients that were admitted to the ICU as previously scheduled. On one hand, normalized postoperative mortality in patients admitted to the ICU exceeded that of the normal-ward patients by more than 10 times. On the other hand, it exceeded the predicted weighed mortality according to the validation cohort from Le Manach et al. by more than 4.5 times. We conclude that prediction of postoperative mortality exclusively based on preoperative variables is uncertain when level of postoperative care is not taken into account.

The POSPOM, as presented by Le Manach et al. in 2016, is based on preoperatively available objective data such as the patient's age, the comorbidities as well as the invasiveness of the scheduled surgical procedure. The patient's individual risk for death following surgery during his hospital stay is normalized to these parameters. Overall mortality in our cohort was slightly increased compared to the weighed mortality reported by the French colleagues. As previously described, the composition of our study population differs from the one described by Le Manach et al., with a general shift towards higher POSPOM scores [15]. Increased mortality of lower-risk patients in our cohort may be explained by the exclusion of outpatient surgery. On the other hand, decreased mortality at higher POSPOM values may indicate more experience with critical patients and with more invasive or emergency procedures in our tertiary care hospital, compared to the generally-distributed study hospital population from Le Manach et al. [10].

The weighed mortality according to the POSPOM allows to compare among different subpopulations, e.g., in abdominal or orthopedic surgery [11,12,14] or differing levels of postoperative care. It is obvious that the level of postoperative care will have a significant impact on the patients' morbidity as well as mortality, since monitoring and therapeutic options on the ICU differ substantially from those on the normal ward. In our cohort, 9.6% of all patients were admitted to ICU during their postsurgical stay, which corresponds to the results of other studies in patient populations of comparable size and composition [16,21]. We subdivided this ICU cohort into two groups according to the time of their admission. The definition of early (up to 24 h following surgery) transfer to the ICU in most cases will include patients with previously scheduled postsurgical admission as well as patients that require continuing advanced monitoring or treatment already commenced in the operating or recovery room or in the post-anesthesia care unit (PACU) [21]. Particularly following major surgery of the elderly or severely ill patient, admission to ICU is considered necessary to recognize and treat critical complications and is therefore routinely scheduled before surgery. Mortality in general is expected to be higher than in normal-ward patients, but usually determined by the preoperative intrinsic risk profile or by intra- or early postoperative complications and lower than in patients being admitted to the ICU later than 24 h following surgery (unplanned admission) [16,21]. In contrast, adverse events following surgery that arise on the normal ward during the further postoperative hospital stay and result in unplanned admission to the ICU are supposed

to be associated with greatly increased mortality [22]. In those cases, the patient's preexisting intrinsic risk profile should play a rather subordinate role for the decision of ICU admission.

As expected, patients scheduled for postsurgical ICU admission revealed higher POSPOM values, indicating increased patient age, more severe comorbidities and more invasive surgical procedures. Therefore, admission to the ICU will already have been planned prior to surgery in many cases [16]. Due to advanced monitoring and therapy, we expected mortality normalized according to POSPOM in this cohort to almost equal that in the normal-ward patients, at least at lower POSPOM values. However, postoperative death was significantly increased, being more than 6 times as frequent as in the normal-ward group and more than 1.5 times as frequent as what was expected from the validation cohort of Le Manach et al. Of note, considerably increased mortality (compared to the patients transferred to the normal ward) occurred even at a low preoperative risk profile, i.e., at POSPOM values of 15. Most likely, this is due to intra- or early postoperative adverse events resulting in transfer to the ICU immediately from the operation or recovery room. Increased mortality in this cohort suggests that even the high level of postoperative care in the ICU cannot fully compensate for any adverse events. However, this confirms recent findings of the International Surgical Outcomes Study (ISOS), questioning patient survival benefit from planned admission to ICU immediately following surgery even after risk adjustment [16]. Results from other studies support this assumption [23].

Postoperative mortality increased even more when patients were admitted to ICU later than 24 h following surgery on an unscheduled basis in order to address postoperative complications arising on the normal ward [22]. POSPOM values of those patients were lower compared to the planned-admission group. This suggests that other factors than solely those defining the POSPOM contribute to postsurgical complications on the normal ward and the need for emergency ICU admission to a high degree. Although the number of patients in this group is rather small, mortality was sharply increased. This may reflect the death rate due to postsurgical adverse events, also termed the '*failure to rescue*' (FTR), and describes the inability of an organization (i.e., a hospital) to adequately deal with postsurgical adverse events arising on the normal ward [24]. FTR is not the occurrence and prevention but the management of postsurgical complications. This phenomenon is not depending on the patient's individual preoperative risk profile (which is the basis for the POSPOM score), but rather on intra- and postoperative events and even more on procedural characteristics concerning the individual hospital. While patient characteristics play a role for the occurrence of adverse events, the death rate due to these postoperative adverse events (the FTR) is determined by hospital characteristics [24]. Therefore, to prevent FTR (not the occurrence of postoperative complications), factors such as more experienced nursing staff, the board certification rate of anesthetists, advanced capabilities for ECG monitoring on normal wards, the presence of Medical Emergency (MET) or Rapid Response Teams (RRT), regular team training and Crew Resource Management (CRM) and established early warning systems such as the National Early Warning Score (NEWS) or instruments to relay information about complications such as SBAR have been shown to be effective [25].

It should be stressed that increased mortality in the ICU subcohorts must not be explained by the fact that the patients were transferred to the ICU (suggesting a hazardous impact of ICU admission and therapy itself). It rather seems to be an effect of the allocation of all patients to either the normal-ward or the ICU cohorts as part of our retrospective analysis. A high proportion of patients experiencing (in the end lethal) intra- or postoperative adverse events is represented in the ICU groups, which are therefore a surrogate for such complications.

Capacities for advanced postoperative monitoring and therapy on the ICU consume substantial human and financial resources and thus are limited [26]. On one hand, unavailability of ICU beds may result in the postponement or even cancelation of surgical interventions [27].

On the other hand, overestimation of the probable need for postoperative ICU admission would result in blocking of bed capacities, which will then not be used in the end. However, as our results reveal, the POSPOM is not appropriate to be used for the preoperative stratification of scheduled postsurgical ICU allocation. In recent years, a number of studies tried to identify risk factors that allow to predict the need for either scheduled or unplanned postoperative ICU admission and therefore facilitate resource allocation [28–35]. Some studies focus on those variables distinguishing patients admitted to normal ward from those admitted to the ICU, including early or even planned admission based on hospital-specific standard procedures [28,30,32]. However, this obviously results in a critical bias due to the ICU admission already scheduled before surgery, even when the results are verified in a validation cohort. Moreover, such approach cannot determine if postoperative care on the ICU will really benefit the patient in terms of individual mortality. In contrast, we used a combination of normalizing the patient's individual risk to the POSPOM score with the retrospective observation of mortality in relation to the level of postoperative care. With this approach we are able to conclude that scoring systems exclusively relying on preoperative parameters ignore factors such as intra- or early postoperative events that influence the need for postsurgical advanced monitoring or therapy to a high degree. Therefore, they are not able to predict the individual benefit of post-operative ICU admission on their own with sufficient accuracy. In their very recent work, Cuijpers et al. drew exactly the same conclusion and state that "*perioperative risk assessment reflecting premorbid physical status [. . .] loses its value when complications occur requiring unplanned ICU admission. Risks [. . .] should be re-assessed based on current clinical condition prior to ICU admission, because outcome prediction is more reliable then.*" [35]. Of course, per-forming a controlled study (which is usually considered to be the gold standard to determine the benefits of an intervention) and allocating patients to either normal ward or ICU on a ran-dom basis would not be possible. However, other authors compared cohorts of patients admit-ted to normal ward with unplanned or emergency admittance to ICU [33] and used propensity score matching [29] or correlated their results with other already established surgi-cal risk prediction scores such as the Emergency Surgery Score (ESS) [34]. Moreover, perfor-mance of risk prediction tools may also vary depending on patient population or surgical subspecialties [28,30,33,35].

An additional postoperative re-scoring, e.g. in the recovery room, would possibly increase the predictive power [36]. Furthermore, although controversially discussed, biomarkers such as Brain Natriuretic Peptide (BNP), cardiac troponin, or blood urea nitrogen (BUN) may help estimating the patient's risk for postoperative mortality and his or her individual benefit from advanced postsurgical monitoring and therapy [14,37–39]. Of note, beside their predictive strength when being assessed preoperatively, postsurgical troponin kinetics or BNP point-of-care testing when patients were reviewed by an MET have also been shown to be able to esti-mate the risk for ICU admission [38,40].

Of course, our study has limitations. Data were generated from a prolonged observation period of 12 years. This was done in order to gain reliable results from a large number of cases. Over the years, changes in personnel (surgeons, anesthetists, nursing staff), improvements in surgical techniques, or learning curves positively modifying surgical skills and procedures occur [41–43], and an influence on the outcomes of individual patients or groups of patients cannot be excluded. However, this should have no substantial impact on the overall results of our analysis, given that the large number of cases mitigates those effects. Since our study is a retrospective single-center cohort observation, selection and information bias cannot be excluded. Moreover, although reflecting frequencies also described by others [16,21], case number in the scheduled and even more in the unscheduled ICU cohort is rather small,

therefore, the results for mortality are not necessarily transferrable to other hospitals or countries and should be re-evaluated by others.

## Conclusions

Prediction of postoperative mortality exclusively based on preoperative variables may be uncertain as it ignores the significance of, e.g., intra- or early postoperative events for the patients' outcome. Therefore, scoring systems such as the POSPOM are limited to predict the individual benefit of postoperative ICU admission.

## Supporting information

**S1 Table. Variables included in the POSPOM score.**
(DOCX)

**S2 Table. Characteristics of patient age and comorbidities.**
(DOCX)

**S3 Table. Distribution of POSPOM score values across surgical disciplines.**
(DOCX)

## Author Contributions

**Conceptualization:** Jan Menzenbach, Maria Wittmann, Tobias Hilbert.

**Data curation:** Jan Menzenbach, Yannik C. Layer, Yonah L. Layer, Tobias Hilbert.

**Formal analysis:** Jan Menzenbach, Yannik C. Layer, Yonah L. Layer, Andreas Mayr, Tobias Hilbert.

**Methodology:** Yannik C. Layer, Tobias Hilbert.

**Project administration:** Tobias Hilbert.

**Resources:** Mark Coburn.

**Supervision:** Mark Coburn, Maria Wittmann, Tobias Hilbert.

**Visualization:** Tobias Hilbert.

**Writing – original draft:** Tobias Hilbert.

**Writing – review & editing:** Jan Menzenbach, Yannik C. Layer, Yonah L. Layer, Andreas Mayr, Mark Coburn, Maria Wittmann.

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
