## [Decision Letter · Decision Letter 0]

15 Jul 2021

PONE-D-21-08078

The level of postoperative care influences mortality prediction by the POSPOM score: a retrospective cohort analysis

PLOS ONE

Dear Dr. Hilbert,

Thank you for submitting your manuscript to PLOS ONE. After careful consideration, we feel that it has merit but does not fully meet PLOS ONE’s publication criteria as it currently stands. Therefore, we invite you to submit a revised version of the manuscript that addresses the points raised during the review process.

We look forward to receiving your revised manuscript.

Kind regards,

Andrea Ballotta

Academic Editor

PLOS ONE

Additional Editor Comments (if provided):

Dear Authors I really apologize for the delay of the answer

but it was really though to find out reviewers available.

Finally we reached a decision: the manuscript should undergo major revision.

Journal Requirements:

2. In the Methods section and the online submission form, please clearly indicate whether the ethics committee (University Hospital Bonn, Bonn, Germany) approve the study protocol.

5. Please upload a new copy of Figures 2 and 3 as the detail is not clear. Please follow the link for more information: https://blogs.plos.org/plos/2019/06/looking-good-tips-for-creating-your-plos-figures-graphics/" https://blogs.plos.org/plos/2019/06/looking-good-tips-for-creating-your-plos-figures-graphics/.

Reviewers' comments:

Reviewer's Responses to Questions

**Comments to the Author**

1. Is the manuscript technically sound, and do the data support the conclusions?

Reviewer #1: Yes

Reviewer #2: Yes

2. Has the statistical analysis been performed appropriately and rigorously? 

Reviewer #1: Yes

Reviewer #2: Yes

3. Have the authors made all data underlying the findings in their manuscript fully available?

Reviewer #1: Yes

Reviewer #2: Yes

4. Is the manuscript presented in an intelligible fashion and written in standard English?

Reviewer #1: Yes

Reviewer #2: Yes

5. Review Comments to the Author

Reviewer #1: The restrospective study describes the applicability of the PosPom score in the German cohort, the paper is well written but it has some limitations and needs major revision.

-In the introduction: the authors wrote “over 400 million surgical interventions can be expected in 2020”, we are in 2021 so it needs some update.

-The observation period is quite long, surgical techniques may have been improved, please comment on that.

-Methods: did you exclude patient undergoing cardiac surgery? Which kind of surgery do you analyze?

-Please specify the variables in the score.

-It could be interesting to compare other preop risk scores

-Do you think POSPOM score is not able to predict the patients really needing ICU? It’s not clear for me.

-Please cite the limits of the retrospective study.

-In the discussion you said: “In contrast, adverse events following surgery that arise on the normal ward during the further postoperative hospital stay and result in unplanned admission to the ICU are supposed to be associated with greatly increased mortality. In those cases, the patient’s preexisting intrinsic risk profile should play a rather subordinate role for the decision of ICU admission.” It seems difficult to me to understand this point, which factors could you assess to prevent or to classify postoperative complications?

-Troponin, BNP and BUN trend could help you after surgery more than before, please comment on that.

Reviewer #2: Thanks for your work.

It is a very interesting analysis that put in the centre the role of postoperative care.

It would be interesting if you can specify the distribution of the surgery in term of specialities.

There are cardiac or neuro surgery in the count?

6. PLOS authors have the option to publish the peer review history of their article (what does this mean?). If published, this will include your full peer review and any attached files.

Reviewer #1: No

Reviewer #2: No

---

## [Author Response · Author response to Decision Letter 0]

22 Jul 2021

Dear Reviewers,

dear members of the Editorial Board of PLOS ONE,

dear Prof. Ballotta,

we thank you for giving us the opportunity to submit a revised version of our manuscript. We have carefully studied the Editor’s and Reviewers’ comments and gratefully acknowledge the helpful suggestions they made to further improve our report. In order to meet their demands, we have made comprehensive modifications within the text, which are further explained in a detailed point-to-point reply. All changes we made to the text are highlighted. Furthermore, we provide you with a clean version of the revised manuscript.

Reviewer #1:

The restrospective study describes the applicability of the PosPom score in the German cohort, the paper is well written but it has some limitations and needs major revision.

In the introduction: the authors wrote “over 400 million surgical interventions can be expected in 2020”, we are in 2021 so it needs some update.

The Reviewer is absolutely right stating that we are in the year 2021 and that this information needs some update. Quite obviously, there is no current data for the exact global volume of surgical procedures in 2020, nor for its estimate in 2021 (either from the WHO, The Lancet Commission on Global Surgery, or from the OECD). Therefore, for 2021, only a cautious estimate can be given on the basis of conservatively assuming a constant growth rate, and we have updated the text of the manuscript accordingly, hoping to meet the Reviewer’s demand. 

The observation period is quite long, surgical techniques may have been improved, please comment on that.

We thank the Reviewer for this critical comment. Data for our analysis were generated from a prolonged observation period of 12 years. This was done in order to gain reliable results from a large number of cases. Over the years, changes in personnel (surgeons, anesthetists, nursing staff), improvements in surgical techniques, or learning curves positively modifying surgical skills and procedures occur1–3, and an influence on the outcomes of individual patients or groups of patients cannot be excluded. However, this should have no substantial impact on the overall results of our analysis, given that the large number of cases mitigates those effects. We now discuss this critical limitation in the corresponding section of our manuscript and added these references to the text. 

Methods: did you exclude patient undergoing cardiac surgery? Which kind of surgery do you analyze?

Please specify the variables in the score.

The Reviewer raises an important point, since this information was actually missing in our manuscript. Patients from all relevant surgery disciplines including cardiac and neurosurgery were included in the POSPOM score. In detail, the disciplines and procedures covered by our study comprised cardiac, urologic, vascular, plastic, ophthalmologic, gynecologic, ear-nose-throat (ENT), orthopedic, transplant, digestive, liver, biliary tract and pancreas, thoracic and neurosurgery. Moreover, other interventions involving anesthesia including endoscopy, neuroradiology and cardiac rhythmology were likewise included. Further variables to derive the score from comprised patient age and comorbidities. This information is now given in the Methods section of our manuscript. Furthermore, all variables included in the score are now shown in detail in Supplemental Table S1. 

It could be interesting to compare other preop risk scores

Do you think POSPOM score is not able to predict the patients really needing ICU? It’s not clear for me.

Thank you for this valuable comment. In recent years, a number of studies tried to identify risk factors that allow to predict the need for either scheduled or unplanned postoperative ICU admission and therefore facilitate resource allocation.4–11 Some studies focus on those variables distinguishing patients admitted to normal ward from those admitted to the ICU, including early or even planned admission based on hospital-specific standard procedures.4,6,8 However, this obviously results in a critical bias due to the ICU admission already scheduled before surgery, even when the results are verified in a validation cohort. Moreover, such approach cannot determine if postoperative care on the ICU will really benefit the patient in terms of individual mortality. In contrast, we used a combination of normalizing the patient’s individual risk to the POSPOM score with the retrospective observation of mortality in relation to the level of postoperative care. With this approach we are able to conclude that scoring systems exclusively relying on preoperative parameters ignore factors such as intra- or early postoperative events that influence the need for postsurgical advanced monitoring or therapy to a high degree. Therefore, in our opinion, such scoring systems are not able to predict the individual benefit of postoperative ICU admission on their own with sufficient accuracy and are, to answer the Reviewer’s question, not able to predict the patients really needing ICU preoperatively. In their very recent work, Cuijpers et al. drew exactly the same conclusion and state that “perioperative risk assessment reflecting premorbid physical status […] loses its value when complications occur requiring unplanned ICU admission. Risks […] should be re-assessed based on current clinical condition prior to ICU admission, because outcome prediction is more reliable then.”.11 Of course, performing a controlled study (which is usually considered to be the gold standard to determine the benefits of an intervention) and allocating patients to either normal ward or ICU on a random basis would not be possible. However, other authors compared cohorts of patients admitted to normal ward with unplanned or emergency admittance to ICU9 and used propensity score matching5 or correlated their results with other already established surgical risk prediction scores such as the Emergency Surgery Score (ESS)10. Moreover, performance of risk prediction tools may also vary depending on patient population or surgical subspecialties.4,6,9,11

All this information together with the respective references has now been included into the Discussion section of our manuscript, hoping this makes our concluding remarks clearer for the reader. We hope to have answered your questions to your satisfaction.

Please cite the limits of the retrospective study.

Of course, our study has limitations. You already mentioned the prolonged observation, making it possible that changes in personnel, improvements in surgical techniques, or learning curves may have influenced the outcomes of individual patients or groups of patients. Furthermore, since our study is a retrospective single-center cohort observation, selection and information bias cannot be excluded. Moreover, although reflecting frequencies also described by others12,13, case number in the scheduled and even more in the unscheduled ICU cohort is rather small, therefore, the results for mortality are not necessarily transferrable to other hospitals or countries and should be re-evaluated by others. These critical limitations are now discussed in the corresponding section of our manuscript. 

In the discussion you said: “In contrast, adverse events following surgery that arise on the normal ward during the further postoperative hospital stay and result in unplanned admission to the ICU are supposed to be associated with greatly increased mortality. In those cases, the patient’s preexisting intrinsic risk profile should play a rather subordinate role for the decision of ICU admission.” It seems difficult to me to understand this point, which factors could you assess to prevent or to classify postoperative complications?

The Reviewer addresses a very important point. Increased mortality in patients admitted to the ICU on an unscheduled basis may reflect the inability of an organization (i.e., a hospital) to adequately deal with postsurgical adverse events arising on the normal ward. This is what is also known as the ‘failure to rescue’ (FTR), and it does not describe the occurrence and prevention but the management of postsurgical complications. This phenomenon is not depending on the patient’s individual preoperative risk profile (which is the basis for the POSPOM score), but rather on intra- and postoperative events and even more on procedural characteristics concerning the individual hospital (see Silber et al.14). While patient characteristics play a role for the occurrence of adverse events, the death rate due to these postoperative adverse events (the FTR) is determined by hospital characteristics. Therefore, to prevent the FTR (not the occurrence of postoperative complications), factors such as more experienced nursing staff, the board certification rate of anesthetists, advanced capabilities for ECG monitoring on normal wards, the presence of Medical Emergency (MET) or Rapid Response Teams (RRT), regular team training and Crew Resource Management (CRM) and established early warning systems such as the National Early Warning Score (NEWS) or instruments to relay information about complications such as SBAR have been shown to be effective.15 We now have modified the Discussion section accordingly to make this point clearer for the reader.

Troponin, BNP and BUN trend could help you after surgery more than before, please comment on that.

This is absolutely right. Biomarkers such as Brain Natriuretic Peptide (BNP), cardiac troponin, or blood urea nitrogen (BUN) may help estimating the patient’s risk for postoperative mortality and his or her individual benefit from advanced postsurgical monitoring and therapy. As the Reviewer states correctly, beside their predictive strength when being assessed preoperatively, postsurgical troponin kinetics or BNP point-of-care testing when patients were reviewed by an MET have also been shown to be able to estimate the risk for ICU admission.16,17 We now have added this information to the text. 

Reviewer #2:

Thanks for your work. It is a very interesting analysis that put in the centre the role of postoperative care.

It would be interesting if you can specify the distribution of the surgery in term of specialities. There are cardiac or neuro surgery in the count?

We thank the Reviewer for raising this interesting point. As already mentioned above, all information on surgical disciplines as well as other variables included in the score (patient age, comorbidities) is now given in the Methods section of our manuscript and in detail in Supplemental Table S1. To follow the Reviewer’s suggestion, the distribution of POSPOM score values across the surgical disciplines as well as the characteristics of patient age and comorbidities are now additionally shown in detail in the Supplemental Tables S2 and S3. We hope to have been able to answer the Reviewer’s questions satisfactorily.

Additional Editor Comments:

Our manuscript now meets all PLOS ONE's style requirements, including those for file naming.

2. In the Methods section and the online submission form, please clearly indicate whether the ethics committee (University Hospital Bonn, Bonn, Germany) approve the study protocol.

As requested, it is now clearly indicated that the ethics committee of the University Hospital Bonn (Bonn, Germany) approved the study protocol.

Data Availability statement: All relevant data sets necessary to replicate the study findings are within the paper and its Supporting Information files.

As recommended, the full ethics statement is now included in the Methods section of our manuscript.

5. Please upload a new copy of Figures 2 and 3 as the detail is not clear.

New high-resolution copies of all figures have been uploaded to make all details clear.

We are convinced that these additional clarifications fulfil the wishes of the Editor and the Reviewers to the fullest extent and that you agree that the changes improved our manuscript and helped to clarify the report. Please do not hesitate to contact us if you should have any more questions. We are very looking forward to your decision.

Thank you very much!

Sincerely,

PD Dr. med. Tobias Hilbert, MD, D.E.S.A. 

References:

1. Vickers, A. J., Cronin, A. M., Masterson, T. A. & Eastham, J. A. How do you tell whether a change in surgical technique leads to a change in outcome? J. Urol. 183, 1510–1514 (2010).

2. Theologie-Lygidakis, N., Chatzidimitriou, K., Tzerbos, F., Kolomvos, N. & Iatrou, I. Development of surgical techniques of secondary osteoplasty in cleft patients following 12 years experience. J. Cranio-Maxillo-fac. Surg. Off. Publ. Eur. Assoc. Cranio-Maxillo-fac. Surg. 42, 839–845 (2014).

3. Acker, S. N., Staulcup, S., Partrick, D. A. & Sømme, S. Evolution of minimally invasive techniques within an academic surgical practice at a single institution. J. Laparoendosc. Adv. Surg. Tech. A 24, 806–810 (2014).

4. Lian, C. et al. Modified paediatric preoperative risk prediction score to predict postoperative ICU admission in children: a retrospective cohort study. BMJ Open 10, e036008 (2020).

5. Knight, J. B., Lebovitz, E. E., Gelzinis, T. A. & Hilmi, I. A. Preoperative risk factors for unexpected postoperative intensive care unit admission: A retrospective case analysis. Anaesth. Crit. Care Pain Med. 37, 571–575 (2018).

6. Khidir, N., El-Matbouly, M., Al Kuwari, M., Gagner, M. & Bashah, M. Incidence, Indications, and Predictive Factors for ICU Admission in Elderly, High-Risk Patients Undergoing Laparoscopic Sleeve Gastrectomy. Obes. Surg. 28, 2603–2608 (2018).

7. Franko, L. R. et al. Clinical Factors Associated With ICU-Specific Care Following Supratentoral Brain Tumor Resection and Validation of a Risk Prediction Score. Crit. Care Med. 46, 1302–1308 (2018).

8. Abrol, N. et al. Preoperative Factors Predicting Admission to the Intensive Care Unit After Kidney Transplantation. Mayo Clin. Proc. Innov. Qual. Outcomes 3, 285–293 (2019).

9. Sukhonthamarn, K., Grosso, M. J., Sherman, M. B., Restrepo, C. & Parvizi, J. Risk Factors for Unplanned Admission to the Intensive Care Unit After Elective Total Joint Arthroplasty. J. Arthroplasty 35, 1937–1940 (2020).

10. Kongkaewpaisan, N. et al. Can the emergency surgery score (ESS) be used as a triage tool predicting the postoperative need for an ICU admission? Am. J. Surg. 217, 24–28 (2019).

11. Cuijpers, A. C. M. et al. Preoperative Risk Assessment: A Poor Predictor of Outcome in Critically ill Elderly with Sepsis After Abdominal Surgery. World J. Surg. 44, 4060–4069 (2020).

12. Kahan, B. C. et al. Critical care admission following elective surgery was not associated with survival benefit: prospective analysis of data from 27 countries. Intensive Care Med. 43, 971–979 (2017).

13. Jerath, A., Laupacis, A., Austin, P. C., Wunsch, H. & Wijeysundera, D. N. Intensive care utilization following major noncardiac surgical procedures in Ontario, Canada: a population-based study. Intensive Care Med. 44, 1427–1435 (2018).

14. Silber, J. H., Williams, S. V., Krakauer, H. & Schwartz, J. S. Hospital and patient characteristics associated with death after surgery. A study of adverse occurrence and failure to rescue. Med. Care 30, 615–629 (1992).

15. Burke, J. R., Downey, C. & Almoudaris, A. M. Failure to Rescue Deteriorating Patients: A Systematic Review of Root Causes and Improvement Strategies. J. Patient Saf. Publish Ahead of Print, (2021).

16. Writing Committee for the VISION Study Investigators et al. Association of Postoperative High-Sensitivity Troponin Levels With Myocardial Injury and 30-Day Mortality Among Patients Undergoing Noncardiac Surgery. JAMA 317, 1642–1651 (2017).

17. Calzavacca, P., Licari, E., Tee, A. & Bellomo, R. Point-of-care testing during medical emergency team activations: a pilot study. Resuscitation 83, 1119–1123 (2012).

---

## [Editor Report · Decision Letter 1]

13 Sep 2021

The level of postoperative care influences mortality prediction by the POSPOM score: a retrospective cohort analysis

PONE-D-21-08078R1

Dear Dr. Hilbert,

We’re pleased to inform you that your manuscript has been judged scientifically suitable for publication and will be formally accepted for publication once it meets all outstanding technical requirements.

Kind regards,

Andrea Ballotta

Academic Editor

PLOS ONE

Additional Editor Comments (optional):

Sorry again for the delay, i apologize as academic editor. After carefully reading i support the green light for your manuscript acceptance
---

## [Editor Report · Acceptance letter]

20 Sep 2021

PONE-D-21-08078R1 

The level of postoperative care influences mortality prediction by the POSPOM score: a retrospective cohort analysis 

Dear Dr. Hilbert:

I'm pleased to inform you that your manuscript has been deemed suitable for publication in PLOS ONE. Congratulations! Your manuscript is now with our production department. 

Kind regards, 

on behalf of

Dr. Andrea Ballotta 

Academic Editor

PLOS ONE